# Economic Growth and Environmental Quality: Analysis of Government Expenditure and the Causal Effect

**DOI:** 10.3390/ijerph191710629

**Published:** 2022-08-26

**Authors:** Mary Donkor, Yusheng Kong, Emmanuel Kwaku Manu, Albert Henry Ntarmah, Florence Appiah-Twum

**Affiliations:** 1School of Finance and Economics, Jiangsu University, 301 Xuefu Road, Jingkou District, Zhenjiang 212013, China; 2School of Management, Jiangsu University, 301 Xuefu Road, Jingkou District, Zhenjiang 212013, China

**Keywords:** economic growth, government finance expenditure, environmental quality, panel quantile regression, NASA republics

## Abstract

Environmental expenditures (EX) are made by the government and industries which are either long-term or short-term investments. The principal target of EX is to eliminate environmental hazards, promote sustainable natural resources, and improve environmental quality (EQ). Thus, this study looks at the impact of economic growth (EG), and government finance expenditure (GEX) on EQ in Northern Africa and Southern Africa (NASA) republics from 2000–2016. The panel quantile regression (PQR) and panel vector autoregressive (PVAR) model in a generalized method of moment framework (GMM) were employed as a framework. The PQR results show that; (i) In Northern republics, GEX had a significant positive effect on EQ at 25%, 50%, and 75% quantiles levels. (ii) In the Southern republics, GEX had a significant negative impact on EQ at 25%. Moreover, the PVAR through the GMM established that EG and GEX are significantly positive while the parameter for CO_2_ is insignificant and negative in the North. However, in the South, GEX and CO_2_ were statistically significant, while EG positively impacts EQ. Lastly, the granger causality report in North indicates uni-directional causation running from LNGEX → LNGDPpc, LNCO_2_ → LNGDPpc, LNFF → LNGEX, and LNFDI → LNGEX. Similarly, there is uni-directional causation in South republics from LNGEX → LNGDPpc, LNCO_2_ → LNGEX, and LNFDI → LNGEX.

## 1. Introduction

Climate change and global warming are causing increasing harm, which has sparked worldwide initiatives and partnerships to combat them through team efforts. The primary cause of global warming is believed to be anthropogenic activities, namely the emissions of greenhouse gases, mostly CO_2_ from the burning of fossil fuels such as oil, gas, and coal, deforestation, and untenable farming approaches which result from the rise in economic activities. Economic activity is identified as the most important predictor of environmental degradation [1,2,3]. Expansion in economic activity means higher average income at the expense of natural resource depletion, hence, environmental degradation [4]. According to the Classification of Environmental Protection Activities (CEPA 2000), environmental expenditure (EX) includes safeguarding sewage management, air and climate, waste management, wildlife protection, R&D, bio-diversity groundwater, and land protection expenses. EX refers to all spending to reduce the environmental hazards caused by the processes above and improve environmental quality EQ [5]. EX is made by both the government and private sectors and can be either short- or long-term expenditures. The primary objectives of these investments are to eliminate environmental harm, promote sustainable natural resource use, safeguard the environment and ameliorate EQ in general [6,7].

According to the European Commission’s 7th Environment Action Programme assessment, increasing EX in public and private sectors is crucial to enhancing EQ. In addition, the initiatives produced within this context and the promotion of green technology are said to reduce the harmful consequences of economic growth (EG) on the environment and climate. Likewise, the reports state that the phenomena of EG and EQ will have a favorable impact on each other, suggesting that EX in public and private sectors might assure environmentally responsible growth by creating employment possibilities [8]. It is critical to developing measures to avoid or mitigate environmental deterioration caused by increased economic activity in light of these developments. As a result, the link between EX and EQ is assessed regarding the size, content, and technical impact of spending. An increase in EX in the context of public expenditures might promote EQ, yet a quantitative increase in public expenditures may cause EQ to degrade due to the scale effect. In particular, government spending encouragement of labor-intensive sectors may help minimize pollution caused by capital-intensive manufacturing processes as in rural electrification using distributed photovoltaic systems (capital intensive) against wood-based home energy (labor-intensive) [9]. The technical impact category of government finance expenditure (GEX) which seeks to stimulate EQ, may be used to assess R&D efforts and investments in the development of technology that does not degrade the environment [10,11]. EX is one effective strategy for halting the steady deterioration of the environment.

Although the environment is seen as a fundamental component of society, it is ignored in terms of GEX due to a lack of urgency in environmental conservation in Africa. In general, most African economies do not prioritize or seek to achieve environmental preservation as a goal, and there have been no uniform laws or regulations defining the role of GEX in environmental preservation. However, the Paris Agreement built on the UNFCC, recently observed (COP26) in Glasgow on 31 October–13 November 2021. One of their goals was to bring all parties together in a common cause to reduce emissions that negatively impact climate [12,13]. If EQ is deemed a luxury public benefit, Furuoka [14] argues that then it should be given appropriate consideration if necessary public goods are provided (water availability, health conditions, food security). This literature review generates the idea that in economies with a substantial quantity of GEX, EQ must be given significant consideration. Increasing GEX enhances environmental legislation, assists in accelerating, and strengthens institutions working to champion EQ [15]. Pollution management and EQ enhancement are dependent on the size of GEX’s environmental footprint and EQ requirements [16,17,18].

Economic development is critical for long-term progress, particularly in emerging countries in NASA. Fincke and Greiner [19] argued that a high rate of EG is a key characteristic of emerging market economies, and the residents’ quality of life and the country’s economic progress are inextricably intertwined. In the broadest sense, it is claimed that increasing economic activity such as production and consumption hastened the destruction of the environment. Pyerina-Carmen [20] posited that lowering greenhouse gases such as CO_2_ and supporting eco-friendly industrial and farming practices are a pivot to long-term growth. Investment in education, infrastructure development, health and medical services, agriculture, and the housing sector, encouraging local and foreign investments, eco-friendly measures, and the proliferation of business-friendly policies, drive countries’ EG. Hence, NASA republics can target these areas to boost the EG of their countries from the grassroots, resulting in growth from the bottom to the top of the economic pyramid. Pollutants such as CO_2_ are produced during industrial production and are one of the most significant causes of environmental damage [1,21,22,23]. Rapid EG increases job prospects and income levels; on the contrary, it reduces societal wellbeing by causing pollution and deterioration of the environment, with natural resource destruction taking the lead. This research contributes to the literature in the following ways:

Firstly, even though several studies on Africa in the area of EG-EQ have been conducted [24,25,26,27,28] these researches have paid less attention to the GEX’s involvement in these outcomes hence, the magnitude of GEX in maintaining environmental quality as countries pursue economic growth have yet not to be established in NASA economies. Thus, this study is the first to use GEX, which adds to the current literature on the growth–environment quality model in NASA economies. This helps to clarify the involvement of the GEX dynamic inside the NASA republics framework for EG-EQ and whether there is a causal relationship between EG, GEX, and EQ in these economies. Secondly, by looking at the topic from the perspective of NASA republics, this research contributes to the regional variations of NASA republics. According to the African literature, many studies focus on Africa, as a region, socio-economic levels, or other sub-groups, with less emphasis on the corresponding sub-regions. The evaluation intends to provide empirical facts for comparison and specific area officials to make policy decisions by looking at the matters from NASA’s perspective. Finally, when causal linkages between EG, GEX, and EQ established from this study will help stakeholders to build effective policies and strategies for EG, GEX, and EQ in NASA republics.

Thirdly, in order to address issues of omitted variables biases (when variables of significant impact are ignored) as suggested by Barreto [29], Clarke, and Wooldridge [30], this study considered fossil fuel and FDI due to their significant contribution to growth rate and environmental degradation. Fossil fuel is the primary energy generation in most NASA economies and is an important determinant of economic growth. Reductions in energy use through fossil fuels affect EG in adverse ways if EG causes energy use. The discovery of Mensah et al. [27] in 22 selected republics in Africa suggested a long-term and short-term bidirectional causal relationship between fossil fuel energy consumption and EG. Moreover, findings from Baz et al. [31], and Gani [32] suggested fossil fuel is a vital contributor to environmental quality degradation. Most multinational companies are trooping into Africa due to its natural resources endowment and hence, FDI has increased in SSA, studies by Ekwueme et al. [33], Vo and Zaman [34], Naz et al. [35], and Chenran et al. [36] unearthed substantial impact of FDI to EG and environmental degradation.

Finally, contributing to the argument in Africa about the link between EG and EQ is far from ended; per the limited research undertaken in this context, Aluko and Ibrahim [37] investigated the link between EG and the technical influence of financial development on EQ. Their study looked at the entire SSA countries whiles Gholipour and Farzanegan [11] and Musah et al. [38] consider MENA and NA, respectively, due to their contribution to global emissions. To fill this research gap, our study is the first of its kind in NASA regions to use GEX from the perspective of sub-regional economies in Africa, which provides a mean-based evaluation. In addition, it allows the series to be observed over time, bolstering the ecological modernization theory, which asserts that innovative planning by several economic managers can assist in distancing EG from ecological destruction [39] and environmental protection activities. Hence this study looks at the relationship existing between EG, GEX, and EQ in NASA economies.

The remaining portions of this report are organized as follows: part two summarizes the literature that guided the investigation, and part three focuses on the study methods. The study’s empirical outcomes are presented in part four, while the report’s conclusions and policy implications are presented in the last section.

## 2. Literature Review

In this review, we evaluate the relationship between EG, GEX, and EQ. For clarity, the literature review is divided into three sections—the overview of EQ-GEX, EG-GEX, and EG-EQ. The next part will go through each nexus in detail, based on current and pertinent information.

### 2.1. Environmental Quality and Government Finance Expenditure

Considering the past research, the chosen nation and region groupings are identified, and many environmental indicators such as ozone (O_3_), CO_2_, SO_2_, PM_10_, water quality, and deforestation are all used. Even though there is no clear consensus on the impact of public and EX on EQ, it is evident that EX has a beneficial impact on EQ in general. However, the overall impact of EX on EQ is equivocal. Using panel data analysis Bernauer and Koubi [40] examined the relationship between GEX and SO_2_ in 42 nations between 1971 and 1996. A positive association between GEX and SO_2_ was discovered in a study that employed numerous political and economic aspects as a control variable. According to this finding, the air quality degrades as the size of the government grows in EG. Lopez et al. [41], using panel data analysis, explored the links between public-goods expenditure and water pollution. The study employed biological oxygen demand as a water pollution indicator from 1980–2005 in 47 countries and SO_2_ and lead levels as air pollution indicators between 1986–1999 in low- and middle-income nations. A statistically significant negative connection was observed between expenditures, public goods, and pollution. In addition, a non-significant link was discovered between total GEX and air and water pollution. The result by Lin et al. [42] from their study on the spillover effect of EX on pollution density for the period of 2005–2009 in 30 regions of China through panel data analysis showed large investment expenditures caused a low level of pollution in high technology growth in the environment with a spillover impact. The spillover impact of EX was statistically significant and beneficial from the findings of this investigation. As a result of the spillover effect, an increase in EX accelerates eco-technological advancement and concluded from the study’s findings that there is a negative link between EX and pollution levels.

Moreover, Halkos and Paizanos [43] used panel data analysis to examine the direct and indirect impacts of GEX on SO_2_ and CO_2_ for 77 nations. While the direct effect of GEX on SO_2_ is statistically significant and negative in the 1980–2000 research, it was insignificant for CO_2_. While GEX had a negative indirect effect in low-income areas, it was a favorable indirect effect in high-income areas. Thus, the direct effect of GEX on SO_2_ and CO_2_ is distinguished by the countries’ income levels. The study of Lopez and Palacios [44] uses panel data analysis to look at the relationships between the total GEX ratio to GDP, the share of good public expenditure in total spending, openness, and the impacts of the energy levy on air pollution levels in Europe’s wealthiest 12 countries from 1995 to 2008. Air pollution indicators included sulfur dioxide (SO_2_), nitrogen dioxide (NO_2_), and oxygen (O_3_). The results showed a significant negative association between overall GEX and public goods spending and SO_2_ and O_3_ emissions. Islam and Lopez [45] evaluated the impact of public goods and social spending on air pollution, including EX by local and centralized governments and state governments, for 51 US areas. The study proxied air pollution as SO_2_, PM_2.5_, and O_3_ from 1983–2008. The results obtained from an unbalanced panel data study suggested that centralized and local GEX reduces air pollution by 0.1 and 0.5 percent for various contaminants. The spending of state administrations was shown to have little impact on pollution. Per the findings, the content of spending was more essential than the size of the public sector, particularly in the case of air pollution. Furthermore, it was asserted that an upsurge in spending aimed at public and social places would improve air quality, although overall, GEX remained the same.

Galinato and Galinato [46] investigated the impact of public goods spending on deforestation and CO_2_ from 1986–1999 for 12 countries. The imbalanced panel data demonstrated a tangible link between public goods spending and deforestation and CO_2_. Nonetheless, the association between total GEX and environmental indicators was significant and positive in the short term but inconsequential in the long term. Gholipour and Farzanegan [11], using panel data analysis, investigated the links between EX and air quality in 14 MENA countries from 1996 to 2015. In the study, PM_10_ and CO_2_ were chosen as air quality measures. The variables of EX and air quality were shown to be cointegrated. However, EX alone was not statistically significant. Nevertheless, it was shown that EX improved air quality by considering organizational structure and governance quality. In a panel data analysis of China’s most polluted seven cities from 2007 to 2015, He et al. [10] discovered a long-term association between EX and air quality index. Per the results, EX and the total air quality index have a favorable relationship. As a result, a 1% increase in EX across the board would result in a 0.051% improvement in air quality. Additionally, assessing individual cities’ results, an increase of 1% in EX would improve air quality by 0.078, 0.035, 0.097, and 0.091 percent in Beijing, Taiyuan, Chongqing, and Lanzhou, respectively. In the cities of Shijiazhuang, Ji’nan, and Urumqi, although, there was no empirical influence of EX on air quality.

### 2.2. Economic Growth and Environmental Quality

Economic growth, as defined by upsurges in GDP, is the primary aim of macroeconomic policymaking, particularly in capitalist nations that emerged after WWII [47]. The “at what cost?” issue is frequently raised since continual GDP expansion is the desired aim. Furthermore, what ecological and natural effects face in the name of EG is a source of concern. Some studies outlined the significant impact of economic expansion on EQ. For example, findings from Asongu et al. [3] studied on criticalities of growth and environmental sustainability suggested a significant positive impact of EG on EQ in African economies. In addition, the study discovered a bidirectional causal linkage between EG and CO_2_ and suggested the region promotes the need for a paradigm shift away from fossil fuels and toward renewables. Studies by Ekwueme et al. [33] on the CO_2_ effect of renewable energy utilization, fiscal development, and FDI in South Africa revealed no significant impact of FDI on CO_2_ but a significant effect of GDP on CO_2_. Ssali et al. [2] and Mensah et al. [27], established a positive influence of EG on SSA’s CO_2_.

Vo and Zaman [34] discovered EG and CO_2_ have a bidirectional relationship. The causality findings substantially corroborate research that revealed a one-way directional link between EG, CO_2_, and FDI, implying that dynamic correlations across metrics within African countries are underappreciated. Naz et al. [35], Shoaib et al. [48], Sunkanmi et al. [49] in Pakistan found a positive and substantial influence of FDI on CO_2_ in the long run. Chenran et al. [36] discovered a significant short-term effect of FDI on CO_2_ emissions in Laos. Using the ARDL approach, Hang and Ucal [50] examined the effect of FDI on CO_2_ for Turkey and concluded that FDI has a considerable influence on CO_2_ emissions. The outcome by Rehman et al. [51] suggested a downward trend in FDI to mitigate the negative consequences of CO_2_ emissions. Furthermore, fluctuating expenditures have non-eco-friendly effects and provide a positive relationship through CO_2_ emissions in the short run. The findings also show that both positive and negative changes in FDI can damage environmental eminence in the long run. Isik et al. [52], Isik et al. [53] findings on EG development and EQ in the US state revealed a negative influence of fossil fuel consumption in Texas, although it is known for oil-producing.

CO_2_ emissions are a significant source of worry across the world. However, the impact on people’s quality of life and the environment varies by area and donates around 70% of total GHG emissions [24,25]. Developed economies traditionally lead global CO_2_ emissions, and emerging economies’ rapidly increasing energy consumption put their aggregate emissions above the developed countries. In 2005, industrialized economies accounted for over 40% of global CO_2_ emissions, developing nations for around 56%, and aviation and maritime transport for the remaining 4% [54]. Africa economically lacks growth, wherein 21st-century electrification is still a problem and sticks to traditional biomass [25]. Although economic growth is an ultimate focus, under the existing interregional trade policy, growing recent economic activity in most African republics will result in considerable growth in FDI, energy usage, GEX, and CO_2_ emissions by 2030 [55].

### 2.3. Economic Growth and Government Finance Expenditure

Keynesians believe that creativity and perilous processing of macroeconomic equilibrium is built on economic indicators such as savings, consumption, national income, and investment embedded in Keynesian growth theories [56]. According to Keynes, in an environment with no market leverage, the government can interfere by enforcing macroeconomic policy to raise aggregate demand for revitalizing activities in an economy—for example, using measures such as increasing GEX and decreasing taxes. Keynesians believe that GEX quickens socio-economic development faster than monetary policy to expand a country’s potential growth. Unquestionably, GEX is fundamental to every country’s stability since the government’s ability to spend, and tax speeds up economic activities and the general market environment. Some scholars investigated the relationship between GEX and EG; however, the outcomes are quite disparate.

Bergh and Karlsson [57] examined government size as the part of tax over GDP and concluded a negative linkage between it and EG. Using panel data from 30 OECD countries from 1995–2014, Zimcik [58] found a negative association between government size and EG. Contrarily, other studies claim a positive relation between GEX and EG. Jiranyakul [59] investigated the impact of GEX on EG in Thailand and confirmed a positive relationship between them. Tatahi et al. [60] explored panel data of 60 countries from 1976 to 2010 and confirmed that achieving high EG is related to high GEX. Chandiol et al. [61] suggested that the government of Pakistan should increase its expenditure on agriculture to improve the national output in agriculture to induce EG. Khac and Tu [62] found that GEX increases along with the development level of nations, and EG positively connects with investment. Even though many varied findings are claimed, the effect of GEX on EG has been proven. Findings by Isik [63] and Manu et al. [64] suggested that an increase in the GEX leads to a rise in EG which subsequently influenced CO_2_ emissions.

The findings reported in the literature above demonstrate that there is widespread debate on the EG-EQ relationship. By presenting GEX into the relationship of EG-EQ, this study aims to determine the influence of EG, GEX on EQ and their causal effect. Thus, by addressing omitted variable issues the study introduced FF and FDI as controlled variables into this relation in the context of NASA republics, thereby expanding the current literature on EG-EQ.

## 3. Data and Variables

This study used panel data of Northern and Southern economies (Algeria, Egypt, Libya, Morocco, Sudan, Tunisia, Angola, Botswana, Namibia, South Africa, Zambia, and Zimbabwe) to research the relationship and causal effect of economic growth, government finance expenditure and environmental quality, from 2000–2016, integrating fossil fuel consumption (FF) and foreign direct investment (FDI). Since environmental issues are becoming more prominent in these economies, attracting the attention of international agencies [65,66]. Information on the factors mentioned above is carried out from the WDI of the World Bank. Table 1 displays the scheme of the index information, while measurements of connection of different factors incorporated in the series data of the study are outlined. EG is essential to sustainable advancement, particularly in emerging economies. Fincke and Greiner [19] argued that a high rate of EG is a crucial characteristic of emerging market economies. The residents’ quality of life and the country’s economic progress are inextricably linked. Pyerina-Carmen [20] recommended that decreasing carbon emissions and encouraging environmentally-friendly industrial and farming techniques are crucial to a long-term growth rate. CO_2_ is used as a proxy for EQ by Halkos and Paizanos [18], Musah, et al. [26], Vo and Zaman [34], Chenran et al. [36], and Yaduma et al. [67] and is a pollutant produced by burning fossil fuels, manufacturing cement, and utilizing solid, liquid, and gas fuels, as well as gas flaring. The research contributes to the discussion of regional disparities in growth and ecological effects and geographic variability in the field [68] and the possible endogeneity problems raised in numerous studies [69]. Certain SSA countries were taken out of the analysis due to a lack of data or inadequate data across the sample period. To increase numerical accuracy, simulate improbability, and eliminate endogeneity glitches, we used a list of potential rheostat indicators identified in the literature to adjust for EG (based on early inquiry assessments). This is in line with the recent study by Musah et al. [26], Ahmad et al. [69], and Bekhet et al. [70], on the crucial role these components play in the growth and the environment.

To comprehend variances in growth and environmental consequences among NASA’s regional blocks, descriptive statistics on a regional basis are presented. This can also be used as a reference point for understanding data disparities. North Africa generates the most significant carbon dioxide (M = 10.388, SD = 1.222), whereas South Africa produces the least (M = 8.955, SD = 1.770), according to Table 2. This indicates that North Africa pollutes the African climate more than South Africa on average. Similarly, North Africa recorded a high growth rate of (M = 8.535, SD = 0.517) than South republics of (M = 8.250, SD = 0.954). Interestingly, North republics that produce high CO_2_ correspond to the economic development rate, and the South that produces less CO_2_ has an average growth out. As illustrated in this summary, NASA’s development and ecological outcomes vary depending on its regional economic and ecological blocks. The authors restricted NASA countries to their different economic and political features to prevent misleading results when investigating this question. The data on kurtosis in the North are mostly mesokurtic (kurtosis values of approximately 3). Except for FDI, the statistics in South Africa have a mesokurtic structure. Apart from CO_2_ and GDPpc, the statistic in Northern Africa is leptokurtic (kurtosis value > 3). The JB and likelihood findings back up Table 2’s informative statistics, which show that the data are normally distributed. In such a distribution, heterogeneous panel data are ideal for researching the issue [27].

### 3.1. Preliminary Analysis

#### 3.1.1. Trend of Variables

Kernel density distribution estimation is used in the study to explore the strength of association amid the parameters. The trend of the study variables for the sub-regional blocks is depicted in Figure 1 and Figure 2. We can observe general trends of each variable in two sub-regions. In the case of Northern, in general, we can observe an upward trend of per capita GDPpc and CO_2_ emissions within the initial state although GDPpc dropped and CO_2_ increased even though there were fluctuations. GEX was gradual but began to rise but got to its peak of one and there was a decline over the period. FDI and FF gradually increased over time but FF had a lot of up and downwards turnover time as compared to FDI which had a smooth rise but decline over time as it hit its peak of 0.6. In the Southern republics, it is observed that GDPpc and CO_2_ fluctuate in the same sequence over time. GEX, FF, and FDI showed a gradual upward increment as well.

#### 3.1.2. Cross-Sectional Dependency and Correlation Analysis

The CD test and the correlation coefficient are reported in Table 3. It is confirmed that both the CD_P-test_ and the CD_LMadj_ test are used to analyze indicators to see if panel data have cross-sectional conditions. The null hypothesis of cross-sectional independence for all the metrics GDP, GEX, CO_2_, FF, and FDI is rejected at 1 percent [71,72]. As a result, panel data with the parameter estimate exhibit cross-sections, according to this hypothesis.

Table 4 displays the significance level of the Pesaran unit root test, Johansen and Westerlund’s long-run co-integration among LNGDPpc, LNGEX, LNCO_2_, LNFF, and LNFDI in Africa. These findings back up the null hypothesis of no co-integration. In addition, the Johansen co-integration test is used, which yields the same results as the Westerlund test. This result suggests that the method used is compelling and intense, implying that the succeeding process is economically substantial and reliable.

### 3.2. Model Estimation

This paper aims to assess the influence of EQ on EG in NASA nations, considering the influence of GEX. To respond to the research quest, five indicators-economic growth, government finance expenditure, carbon emission, fossil fuel and foreign direct investment were evaluated. According to the growth impact, GDPpc improves EG by encouraging investment activities, which promote per capita income, decline energy concentration, and expands the environment by reducing CO_2_ [73,74]. As a result, we use the modified Cobb–Douglas production function in this study:(1)$$Y_{it}=A_{it}{}^{*}f\left(K_{it,}L_{it}\right)$$
where *Y_it_* is the output of countries *i* at time *t*, *A_it_* is the Hicks technological factor, *K_it_* signifies capital quantified as the *PKS*, and *L_it_* is the labor. The Cobb–Douglas is a production approach that is widely used as an apt tool to find relations between production and economic factors:(2)$$Y_{it}=A_{it}K_{it}{}^{\infty }L_{it}{}^{\beta }$$
where ∞ and β are the elasticities of *Y* vis-à-vis *L* and *PKS*. Therefore, from the extended Cobb–Douglas production function, it is assumed that the levels of government finance expenditure [17,18,75] capital *K_it_* = *GEX_it_* in the model, which includes CO_2_ proxying EQ, which is extended as:(3)$$Y_{it}=A_{it}GEX_{it}{}^{\delta }L_{it}{}^{\beta }CO_{2}{{}_{it}}^{\gamma }$$
where *CO*_2*it*_ and *GEX_it_* of country *i* and time *t* in years. Here δ and γ are the elasticities of output. Taking the natural logarithm and dividing both sides of Equation (3), we have panel data under per capita terms by the population. Meanwhile, the effect of labor is kept unchanged, and we add fossil fuel and FDI as control variables. We now get a log-linear version of the production function as follows:(4)$$\ln Y_{it}=\ln A_{it}+\delta GEX_{it}+\gamma CO_{2}{}_{it}+\sigma FF_{it}+\tau FDI_{it}$$
where ln*A_it_* = $\beta_{0}+\varepsilon_{it}\;$with *β*_0_ measuring the mean efficiency level across cross-sections and overtime while $\varepsilon_{it}$ is the country-specific deviations from the mean. Equation (4) can be written in the following form:(5)$$\ln Y_{it}=\beta_{0}+\delta GEX_{it}+\gamma CO_{2}{}_{it}+\sigma FF_{it}+\tau FDI_{it}+\varepsilon_{it}$$

In the previous model, *Y_i_**_t_* is the response variable representing *GDPpc_it_*, *GEX_it_*, *CO*_2*it*_ *FF_it_*, and *FDI_it_* as explanatory variables. Therefore, the study attempts to calculate the *PK* corresponding to the individual countries within the panel. The linear relation between EG and EQ can be expressed as follows:(6)$$\ln GDPpc_{it}\;=\;f\left(\ln GEX_{it},\ln CO_{2}{}_{it},\ln FF_{it},\ln FDI_{it}\right)$$

The forecast model’s empirical design is affirmed as:(7)$$\ln GDPpc_{it}=\beta_{0}+\beta_{1}GEX_{it}+\beta_{2}CO_{2}{}_{it}+\beta_{3}FF_{it}+\beta_{4}FDI_{it}+\varepsilon_{it}$$
(8)$$\begin{array}{l} \Delta \ln GDPpc_{it}\;=\;\mu_{1i}\;+\;\sum_{j=1}^{p}\beta_{1j}\Delta \ln GEX_{it-j}+\;\sum_{j=1}^{p}\phi_{1j}\Delta \ln CO_{2}{}_{it-j}+\;\sum_{j=1}^{p}\varphi_{1j}\Delta \ln FF_{it-j}\\ \;\;\;\;\;\;\;\;\;\;\;\;\;\;\;\;\;\;\;\;\;\;+\sum_{j=1}^{p}\gamma \Delta \ln FDI+\alpha_{1i}\;+\;\delta_{1t}+\;\varepsilon_{1it} \end{array}$$

Environmental research frequently uses quantile regression, which has recently emerged as a major area of study because the normalcy evaluation uncovers information that isn’t commonly recognized, the model predisposes OLS by restricting such measures. In such instances, the regression takes into account the findings of various percentiles of dispersion and varied responses. Many contacts to be regulated by OLS and another outmoded econometric approach were overlooked by regression projection for the intensity of the irregularity and the capacity to tighten greatly. We employ the PQR approach of Fan et al. [76] which is used to assess outcomes based on present conditions that exist irrespective of everyone’s capacity or to create presumptions circulation [77,78,79]. The controlled percentile was evaluated as $Q_{Y}\left(\tau |X\right)$ a situation quantified in the form:(9)$$Q_{\tau }\left(\ln GDPpc_{it}\right)=\alpha_{\tau }\;+\;\beta_{1\tau }\ln GEX_{it}+\;\beta_{2\tau }\ln CO_{2}{}_{it}+\;\beta_{3\tau }\ln FF_{it}+\;\beta_{4\tau }\ln FDI_{it}+\varepsilon_{it}$$

Equation (10) represents the PQR Equation of *GDPpc* depending on fixed effects ($\alpha_{\tau }$), *GEX*, *CO*_2_, *FF*, and *FDI*, where *Q_τ_* indicates the QR components of the τth differential point that posits the differential point for the predictor variables. To address endogeneity issues, the lagged evaluated measures were employed in Equation (10) [39,80].

Again, Pedroni, Kao, and Johansen’s co-integration show that variables are integrated in the order I(0) and I(1), and the PVAR model is employed. The most crucial source of macroeconomic dynamics in an open economy is endogenous and exogenous shocks, which are modeled by the PVAR. The PVAR model has no bias against any particular theories of economic expansion. The PVAR model, in contrast to VAR, adds to the dynamic heterogeneity of our data, enhancing measurement of consistency and coherence, particularly where there is heterogeneity in GDP per capita, CO_2_ emissions, GEX, FF, and FDI among NASA republics. Conferring to Love and Zicchino [80], the procedure for PVAR is as follows:(10)$$Y_{it}\;=\;\mu_{i}\;+\;A(L)Y_{it}+\;\alpha_{i}\;+\;\delta_{t}+\;\varepsilon_{it}$$

*i =* 1, 2, 3… N *t =* 1, 2, 3…T.

## 4. Results and Discussion

The quantile regression results in Table 5 demonstrate that conditioning on other determinants of GEX on EQ had a substantial positive impact at all levels of quantiles in the Northern republics. However, the marginal effect rises at 1% from lower to higher quantile. As a result, if all other parameters remain equal, a 1% rise in the GEX will result in an 11.4 percent increase in EQ spending, from a low growth rate of (25th quantile) to a high growth rate of (25th quantile) (75th quantile). According to the findings, nations on a high-growth road are more likely to benefit from boosting the GEX than those on a low-growth path. CO_2_ emission has no significant impact on growth and EQ. FF consumption has a positive and significant impact from the lower quantile to the highest quantile.

Interestingly, an increase in FF increases EQ by 34.5%, 6.9%, and 34.9%. Last, FDI has a positive and significant effect on EQ. All other things being equal FDI increases from the lower quantile to the highest quantile. In the South, GEX had a negative and significant impact on EQ, meaning that GEX on the environment is minimal and has no impact on EQ. The other determinants (CO_2_, FF, and FDI) of EQ in the South are reported to have no impact on the environment.

We use a graphical methodology to emphasize the marginal effects of GEX on the EG of economies at different growing levels to prove further the acceptability of employing the quantile regression method over the OLS method. Using quantile regression and OLS results, the effects of GEX efficiency on EG are visually illustrated in Figure 3 and Figure 4. The coefficients surrounding the quantile estimates (green line or a gray area in confidence interval term) fluctuate dramatically with the diverse quantiles of EG. In contrast, the OLS approach (dotted line) remains static in the chosen quantiles. Because the quantile regression evaluations provide varying outcomes at different distributive locations on the graph, it is evident that utilizing OLS estimations to create a constant estimate for all economies would result in biased estimations because it does not justify outliers, as seen in Figure 3 and Figure 4. As a result, the quantile graph proves the suitability of the quantile regression approach over OLS.

### 4.1. PVAR Results

The optimum lag for PVAR estimations, per Andrews and Lu [81], was among the selected lag best three criteria: MQIC, MBIC, and MAIC. Hence, we employed the GMM approach to applying the first-order lag PVAR model in its stationary version, as suggested by Love and Zicchino [82] and adopted by Andrews and Lu [81]. The PVAR results are shown in Table 6. In North Africa, the parameters of EG are positive and significant, while the parameter for CO_2_ is negative and insignificant. Meaning that expansion in these countries’ economies increases their CO_2_ emissions, thus, decreasing EQ, as stated by previous studies by Ssali et al. [2] and Mensah et al. [27], who discovered that EG had a substantial influence on SSA’s CO_2_ levels.

Interestingly, in South Africa, the parameter of EG is positive and significant, while the parameter for CO_2_ is significant and negative. This can be explained as expansion in nations’ economies grows, and increases CO_2_, hence improving EQ, but the growth of improvement does not affect EQ. This supports the findings of Vo and Zaman [34] that EG is positively related to CO_2_ and FDI, suggesting that GEX improves EQ in African countries. Similarly, the parameter of GEX is positive and significant in the North, which explains that GEX improves EG and EQ [10,44,46]. The parameter of GEX in the South had a positive and insignificant outcome on EG and EQ. FDI in the Northern republics positively and significantly affected EG and EQ. Moreover, the parameter of FDI had a positive and insignificant outcome on growth and the environment. Our outcomes support references [22,35,36,49,83], revealing FDI had a substantial impact on CO_2_ emissions. FDI has historically been a trademark of globalization, assisting in transforming whole companies, cities, industries, and economies. FDI has been a tried and tested method for capital, technology, and transfer of better life throughout the world for decades. Aside from increasing GDP through diversified businesses investment portfolios expansion and lowering poverty, it stimulates innovations, knowledge sharing, modernized business practices, and environmental protection. Nonetheless, the governments of host nations have been drawn to the negative impact of FDI on their environment. They have opted for high-quality investment that tackles EQ, climate change, and advanced sustainable development.

### 4.2. Variance Decomposition

The VD is used to determine how much of the dependent indicator’s variation over time can be explained by its latency and other descriptive indicators. The findings from VD are summarized in Table 7. Because most of the components have the most control over the others, we comprehend the results for the tenth period of this study. In North Africa, evidence from Table 7 shows that LNGDPpc is explained by another determinant by 17%, 17%, 39%, 19.7, and 8.3% for the future variations due to shocks in LNGEX, LNCO_2_ LNFF, and LNFDI. Moreover, 5.1%, 38.8%, 46.3%, 8.6%, and 1% of future fluctuation in LNGFE are due to shocks in LNGDPpc, LNFF, and LNFDI. Moreover, 5.4%, 3.9%, 82.1%, 6.8%, and 1.9% of future fluctuation in LNFF are due to shocks in LNGDPpc, LNGEX, LNCO_2_, and LNFDI. Furthermore, 9%, 7.9%, 2.2%, 73.5%, and 6.9% of future fluctuation in LNCO_2_ are due to shocks in LNGDPpc, GEX, LNFF, and LNFDI. Lastly, 6.3%, 45.2%, 25.4%, 2.0%, and 3.1% of future fluctuation in LNFDI are due to shocks in LNGDPpc, LNCO_2_, LNFF, and LNFDI, which support. Muhammed and Khan [84] concluded the positive influence of FDI on CO_2_ for Asian economies. Economic growth is comparatively higher in value than SE. Economic growth in the West confirms the ARDL results in Table 5 where the impact of EG is greater than renewal energy accounting for (25.7%, and 85.4%). This implies that when EG increases, the CO_2_ and FF impact is substantially minimal and thus improves the environment, and requires less innovation and technology to control and manage energy renewal which supports Ssali et al. [2] and Mensah et al. [27] who established positive impact of EG on SSA’s CO_2_.

In the South, the results show that 66.5%, 5.4%, 21.2%, 0.2%, and 6.7% of impending fluctuations in LNGDPpc are due to shocks in LNCO_2_ LNFF LNGEX, and LNFDI. In addition, 1.3% and 98.7% of future fluctuation in LNGEX are due to shocks in LNGDPpc. Moreover, 94.2% and 5.7% of the future variation in LNFF are due to shocks in LNCO_2_. Furthermore, 62.3%, 7.4%, 7.5%, 19.4%, and 3.4% of future fluctuation in LNFDI are due to shocks in LNGDPpc, LNCO_2_, LNFF, and LNFDI [34,36].

### 4.3. Impulse Response Analysis (IRA)

The (IRA) is used in this study to look at how LNGDPpc, LNGEX, LNCO_2_, LNFF, and LNFDI react to random changes in each other that are not clarified by the Granger causatives test. The (IRA) avoids the orthogonal pitfalls that out-of-sample Granger causality tests are prone to. Figure 5 and Figure 6 show the LNGDP (IRA) to Cholesky One SD. Innovations in other factors. In North Africa, confirmation from Figure 5 displays that the reaction of LNGDP to innovation is significant within the 4-period limit. The initial response of LNGDPpc to LNGEX, LNCO_2_, LNFF, and LNFDI is positive and significant, which supports [35] in Pakistan, disclosed a positive effect from FDI to CO_2_ in the long run.

However, a one SD shock to LNCO_2_, LNFF, and LNFDI to the 1-period horizon increases, but LNFDI decreases. Moreover, a one SD shock to LNCO_2_ peaks at the 4-period limit and upsurges at a constant rate afterward, and a one SD shock to LNFDI peaks at the 5-period limit and surges progressively with time. A one SD shock to LNFF peaks at the 1-period horizon and regularly rises with time. Finally, a one SD shock to FDI peaks at the 2-period horizon and increases progressively with time.

### 4.4. Granger Causality Test

The Granger test investigates the causality among LNGDPpc, LNGEX, LNCO_2_, LNFF, and LNFDI of the associations amongst these parameters [85,86]. Table 8 summarizes the Granger test. The H:0 that LNCO_2_ does not → LNGDPpc, LNGDPpc does not → LNCO_2_, LNFF does not → LNGDPpc, LNGDPpc does not → LNFF, LNFDI does not → LNGDPpc, and LNGDPpc does not → LNFDI is rejected at the 1%, 5%, and 10% significance level. In North Africa, there is uni-directional causation running from LNGEX → LNGDPpc, LNCO_2_ → LNGDPpc, LNFDI → LNGDPpc, LNCO_2_ → LNGEX, LNFF → LNGEX, and LNFDI → LNGEX, and the findings support studies that found a one-way directional relationship between EG, CO_2_, and FDI [24,83]. In South republics, uni-directional causation runs from LNGEX → LNGDPpc, LNFF → GDPpc, LNFDI → LNGDPpc, LNCO_2_ → LNGEX, and LNFDI → LNGEX [31,32,87].

The heterogeneous Granger causalities are evident in the short run. The panel (a) of Figure 7 shows unidirectional causalities running from GEX, CO_2_, FDI and FF to GDPpc, and from CO_2_ to GEX, in Northern republics. In summary, these causalities imply that GEX and FF deployment can serve as a catalyst for regional FDI-fueled sustained GDPpc growth. Nevertheless, policymakers’ biggest obstacle is finding the ideal balance between GEX and CO_2_ emissions [88] vv. However, panel (b) of Figure 7 shows unidirectional causalities running from GEX, CO_2_, FF and FDI to GDPpc, from GEX to CO_2_, from FF to FDI, and from FDI to GEX in Southern Africa. In general, these causalities suggest that the region’s GDPpc emanates from GEX, FDI, and FF. CO_2_ should not be overlooked as FDI and FF increase in the region to improve the growth rate. Again, obtaining a balance between GEX and CO_2_ is equally important to the growth of these republics.

## 5. Conclusions

In recent years, there has been a focus on enacting expansionary macroeconomic policies to mitigate the negative consequences of economic crises. During these years, GEX in many nations has expanded dramatically, influencing economic performance and other welfare metrics. Although improving EQ is not a primary aim of fiscal policy, owing to a lack of public approval, it is vital to analyze the implications of these policies on the effectiveness of environmental legislation and their possible influence on the level of pollution. In this regard, this research empirically examines the relationship between EG and GEX on EQ by evaluating the channels that support the relationship. We investigate the supposition that EG and EQ improve the mitigating impact of GEX on air pollution in NASA regions in particular. Using PVAR and quantile regression, data from a panel of nations from 2000 to 2016 were used to estimate the drivers of LNGDPpc, LNGEX, LNCO_2_, LNFF, and LNFDI. The following are the findings of this investigation:

The quantile findings in the North republics show that GEX had a significant positive effect at all levels of quantiles. However, the marginal effect increases by 1% from lower to higher quantile. In the South, GEX significantly impacted EQ negatively. The other determinants (FF, and FDI) of EQ in the South were reported to have no impact on the environment.

In addition, according to the PVAR through the GMM style in North Africa, economic growth is positive and significant, while the parameter for CO_2_ is insignificant and negative. Meaning that the economies of these countries expand with CO_2_, thus declining EQ, and in Southern Africa, economic growth is positive and significant while the parameter for CO_2_ is significant and negative. This implies that as these countries’ economies expand with CO_2_ emissions, improving EQ.

Moreover, the IRA in North Africa reports that other determinants explain LNGDPpc by 17%, 17%, 39%, 19.7, and 8.3% for the future variations due to shocks in LNGEX, LNCO_2_ LNFF, and LNFDI. Moreover, 5.1%, 38.8%, 46.3%, 8.6%, and 1% of future fluctuation in LNGEX are due to shocks in LNGDPpc, LNFF, LNFDI. In the South, the results show that 66.5%, 5.4%, 21.2%, 0.2%, and 6.7% of future fluctuations in LNGDPpc are due to shocks in LNCO_2_ LNFF LNGEX, LNFDI.

Lastly, in the granger causality report In North Africa, there is uni-directional causation running from LNGEX → LNGDPpc, LNCO_2_ → LNGDPpc, LNFDI → LNGDPpc, LNCO_2_ → LNGEX, LNFF → LNGEX, and LNFDI → LNGEX, the granger outcomes support studies of [24,83] who discovered a one-way relationship between EG, CO_2_, and FDI. Similarly, in South republics, unidirectional causation runs from LNGEX → LNCGDPpc, LNFF → GDPpc, LNFDI → LNGDPpc, LNCO_2_ → LNGEX, and LNFDI → LNGEX.

### 5.1. Policy Implication

This study provides confidence to macroeconomic policymakers in NASA republics that increasing fiscal spending is not harmful to EQ and may significantly relieve air pollution, particularly in industrialized nations. As a result, fiscal expenditure might be helpful in extra efforts to reduce air pollution, making them readily and less expensive. The impact of GEX on EQ in developing countries can be bolstered by reducing policy flaws such as the protection of industry and energy subsidies and enforcing property rights over natural resources, which can help internalize environmental outlays the sources of pollution [89]. Furthermore, policymakers in sub-regional economies in NASA should stimulate GEX by encouraging green technologies. Thus, policymakers should focus on GEX to stimulate economic growth and improve the environment (especially the regional blocs including ECOWAS, CEMAC, EAC, and SADC). The governments in the various economic syndicates should develop regional policies and incentives to help businesses adapt to environmentally sustainable production processes, which will help ease the government’s pressures to improve EQ.

Again, various additional elements that may impact EQ should be addressed when formulating fiscal policies, such as the composition of GEX and the cumulative effect of each policy on economic development and government debt sustainability. Finally, it is critical to construct NASA areas’ sub-regional and macroeconomic policies so that EQ is improved. Increasing EQ in the nations studied would require changing the composition of GEX rather than the size. As a result, policymakers should raise the proportion of EX in GEX. Expanding public-goods spending and replacing private-goods spending, for example, might boost EQ by reducing reliance on natural capital, such as energy consumption. Transferring funds to investments in R&D that promote eco-friendly technologies, such as the utilization of renewable energy sources, will significantly reduce the number of pollution-spreading technologies with negative environmental consequences while enhancing EQ should be considered by governments of NASA regions. The recent COVID-19 pandemic has opened most countries up to newer governmental policies which are geared toward self-reliant and cleaner technologies which enhance EQ. Future research could look into using other environmental indicators to investigate the topic.

### 5.2. Limitation of the Study

Two significant limitations were faced in this study and need attention for future studies. First, the study period was restricted to 2000–2016 due to data limitations for the variables and could not include other economic indicators from the selected regions, which was far shorter than what researchers originally anticipated. When these data are completely made accessible, researchers advise that future research should take longer study periods into account. Additionally, because NASA republics differ in terms of their geographic location, culture, political system, and economic structure, the findings of this study cannot be extrapolated to other African nations or the entire world. Therefore, extrapolating data that solely apply to NASA republics to other countries might result in false conclusions. Despite the aforementioned restrictions, the study’s objectives were met.

## Figures and Tables

**Figure 1 ijerph-19-10629-f001:**
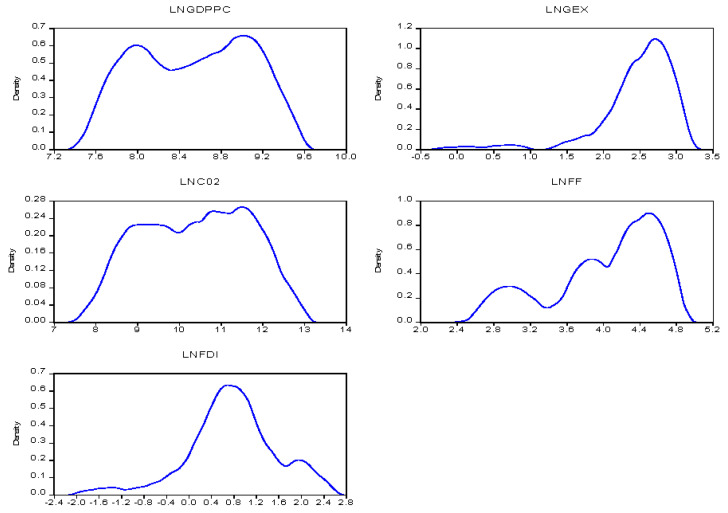
The trend of the indicators-North.

**Figure 2 ijerph-19-10629-f002:**
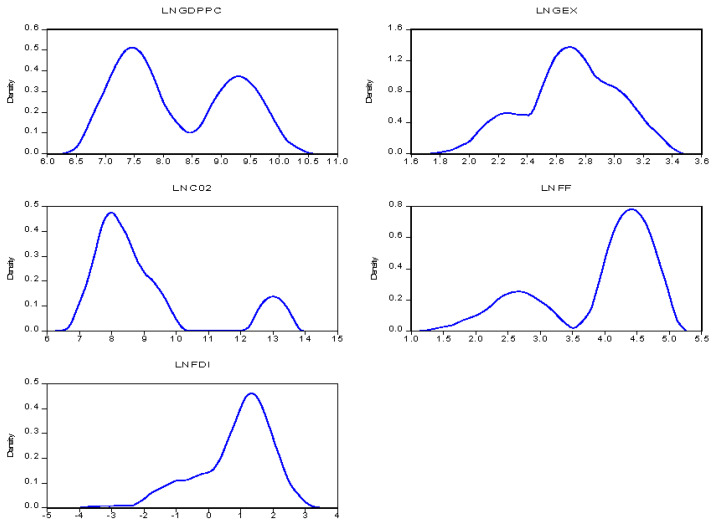
The trend of the indicators-South.

**Figure 3 ijerph-19-10629-f003:**
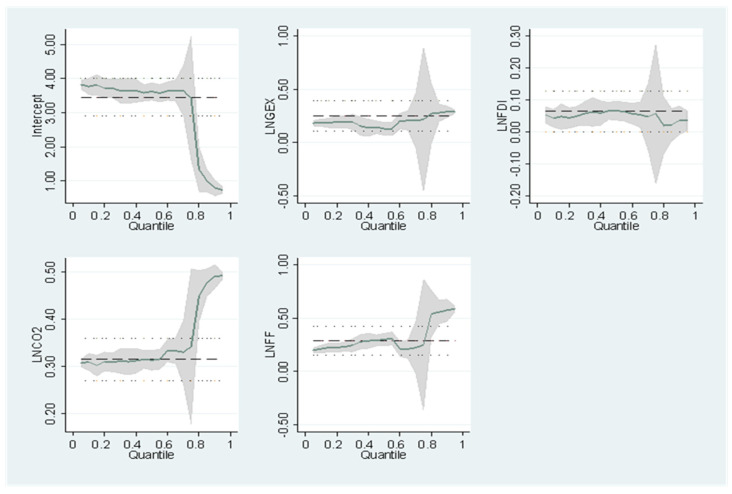
PQR-North.

**Figure 4 ijerph-19-10629-f004:**
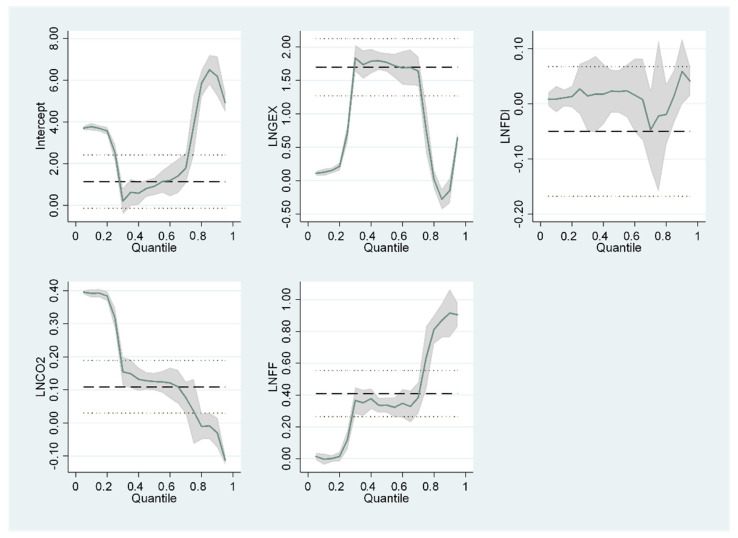
PQR-South.

**Figure 5 ijerph-19-10629-f005:**
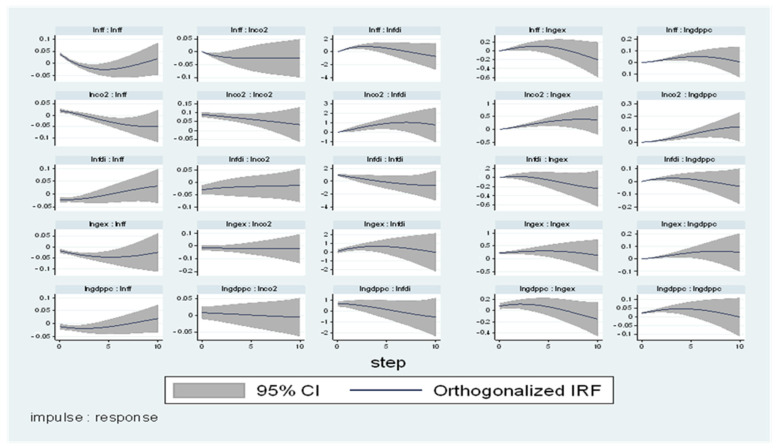
IRF-North.

**Figure 6 ijerph-19-10629-f006:**
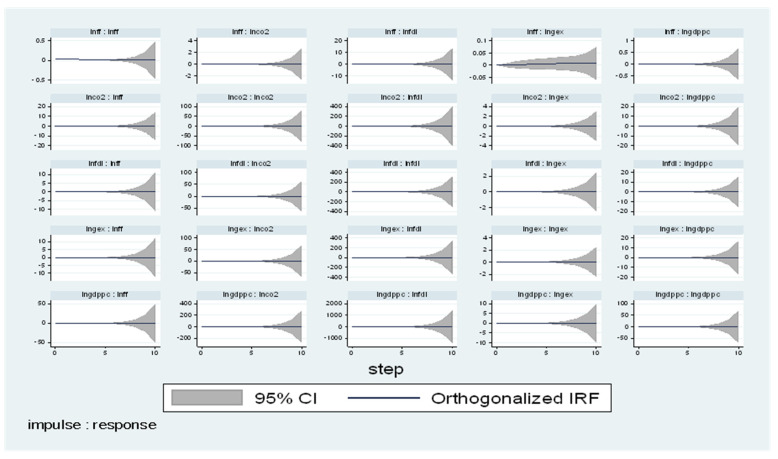
IRF-South.

**Figure 7 ijerph-19-10629-f007:**
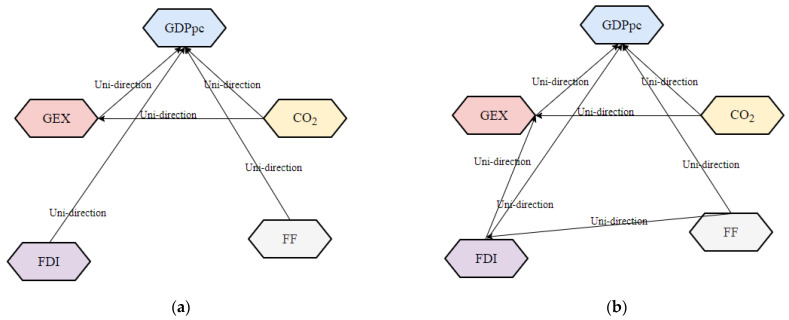
The direction of causality. (**a**) shows unidirectional causalities running from GEX, CO_2_, FDI and FF to GDPpc, and from CO_2_ to GEX, in Northern republics; (**b**) shows unidirectional causalities running from GEX, CO_2_, FF and FDI to GDPpc, from GEX to CO_2_, from FF to FDI, and from FDI to GEX in Southern Africa.

**Table 1 ijerph-19-10629-t001:** Variable definition.

Var.	Indicators	Index (Code)	Source
Economic growth	GDP_PC_	The aggregate gross value added to the economy by all domestic manufacturers, plus any product tariffs, minus any subsidies not included in the product value. It is estimated without considering the depreciation of manufactured assets or natural resource depletion and deterioration.	WDI (2020)
Government finance expenditure	GEX	Transfer payments, which include wage transfers (pension, social benefits) and capital transfers, along with expenditure, such as government expenditure and investment	WDI (2020)
Environmental quality (EQ)	CO_2_ Fossil fuel	Pollutants are produced by the combustion of fossil fuels, the manufacture of cement, the use of solid, liquid, and gaseous fuels, and also gas flaring.	World Bank (2020)
Foreign direct investment	FDI	Investing in commercial interests in a different country by people or companies in another country. In other terms, FDI is an investment by a foreign entity in the form of controlling ownership in a firm in another nation. Foreign direct investment (% of GDP)	World Bank (2020)

**Table 2 ijerph-19-10629-t002:** Descriptive statistics.

	North					South				
	GDPpc	GEX	CO_2_	FF	FDI	GDPpc	GEX	CO_2_	FF	FDI
Mean	8.535	2.461	10.388	4.026	0.763	8.250	2.697	8.955	3.889	0.831
Median	8.569	2.556	10.350	4.205	0.765	7.814	2.683	8.273	4.386	1.160
Maximum	9.284	3.019	12.288	4.589	2.253	9.882	3.267	13.128	4.597	2.716
Minimum	7.723	−0.051	8.278	2.763	−1.660	6.956	1.937	7.119	1.801	−3.218
Std. Dev.	0.517	0.539	1.222	0.575	0.784	0.954	0.308	1.770	0.862	1.127
Skewness	−0.074	−2.359	−0.078	−0.880	−0.472	0.273	−0.264	1.578	−1.042	−1.038
Kurtosis	1.569	9.950	1.731	2.452	3.942	1.394	2.506	4.146	2.432	3.765
Jarque–Bera	8.794	299.962	6.952	14.451	7.569	14.25	2.595	55.927	23.151	24.286
Probability	0.012	0.000	0.031	0.000	0.022	0.000	0.273	0.000	0.000	0.000
Sum Sq. Dev.	26.969	29.403	150.982	33.404	62.167	107.470	11.248	369.7544	87.831	150.064
Observations	102	102	102	102	102	119	119	119	119	119

**Table 3 ijerph-19-10629-t003:** Cross-sectional dependency and correlation analysis.

**Cross-Sectional Dependence Test**										
	**North**					**South**				
**Var.**	**CD_P_ Test**	***p*-Value**	**CD_LMadj_ Test**	***p*-Value**		**CD_P_-Test**	***p*-Value**	**CD_LMadj_ Test**	***p*-Value**	
GDPpc	15.124 ***	0.000	38.880 ***	0.000		12.879 ***	0.000	33.100 ***	0.000	
GEX	3.035 ***	0.002	5.775 ***	0.000		−1.760 **	0.078	8.780 ***	0.000	
CO_2_	10.710 ***	0.000	21.917 ***	0.000		15.927 ***	0.000	36.129 ***	0.000	
FF	2.603 ***	0.001	4.361 ***	0.000		5.637 ***	0.000	7.823 ***	0.000	
FDI	4.395 ***	0.000	308.997 ***	0.000		4.239 ***	0.000	3.329 ***	0.000	
**Correlation-North**						**Correlation-South**				
LNGDPpc	1					1				
LNGEX	0.305	1				0.586	1			
LNCO_2_	0.693	−0.244	1			0.435	0.366	1		
LNFF	0.5724	0.708	0.084	1		0.307	−0.107	0.033	1	
LNFDI	0.0146	0.021	−0.147	0.067	1	−0.095	−0.056	−0.345	0.175	1

Note: ***, ** shows the rejection of the null hypothesis at 1% and 5% significance: the CD_P_—the test of Pesaran 2004 and CD_LMadj_.

**Table 4 ijerph-19-10629-t004:** Cointegration test.

	**North**				**South**			
	**Pedroni**							
	**Statistic**	**Prob.**	**Statistic**	**Prob.**	**Statistic**	**Prob.**	**Statistic**	**Prob.**
Panel v-statistic	−0.663	0.746	0.191	0.424	−0.308	0.621	−0.104	0.541
Panel rho-statistic	0.677	0.751	0.732	0.768	0.502	0.692	1.164	0.877
Panel PP-statistic	−1.737 **	0.041	−2.115	0.017	−3.715 ***	0.000	−2.442 **	0.007
Panel ADF-statistic	−0.288 **	0.086	−0.645	0.259	−2.256 **	0.012	−3.568 ***	0.000
Alternative hypothesis: individual AR coefs. (between-dimension)								
	Statistic	Prob.			Statistic	Prob.		
Group rho-statistic	1.694	0.954			2.397	0.991		
Group PP-statistic	−5.363 **	0.000			−2.010	0.022		
Group ADF-statistic	−0.683 **	0.047			−4.104 ***	0.000		
**Kao**					**Kao**			
ADF	t-Statistic	Prob.			ADF	t-Statistic	Prob.	
	−1.561 *	0.059				−0.928 **	0.076	
Johansen					Johansen			
Hypothesized	Fisher Stat. *		Fisher Stat. *	Hypothesized	Fisher Stat. *			
No. of CE (s)	(from trace test)	Prob.	from the max-eigen test)	Prob.				
None	110.5	0.000	110.5	0.000	77.84	0.000	77.84	0.000
At most 1	190.7	0.000	140.0	0.000	186.5	0.000	111.1	0.000
At most 2	95.67	0.000	57.01	0.000	105.7	0.000	70.05	0.000
At most 3	57.26	0.000	50.56	0.000	55.69	0.000	39.44	0.000
At most 4	23.94	0.020	23.94	0.020	44.27	0.000	44.27	0.000

Note: *, **, and ***, represent the statistical significance at 1% and 5%, and 10% levels, this represents the rejection of the H:0 at 1% and 5%, and 10% levels of significance.

**Table 5 ijerph-19-10629-t005:** Quantile regression.

		North				South	
	ALL	25%	50%	75%	25%	50%	75%
LNGEX	0.535 *** (0.092)	0.114 *** (0.037)	0.097 *** (0.026)	0.081 * (0.033)	−0.215 * (0.403)	−0.118 (5.927)	−0.046 (10.069)
LNCO_2_	0.247 *** (0.025)	−0.096 (0.071)	−0.068 (0.050)	−0.042 (0.063)	−0.0228 (0.357)	0.044 (5.243)	0.093 (8.906)
LNFF	0.276 *** (0.056)	0.345 *** (0.098)	0.347 *** (0.069)	0.349 *** (0.088)	0.202 (0.417)	0.131 (6.115)	0.077 (10.387)
LNFDI	0.030 (0.043)	0.018 * (0.009)	0.014 * (0.007)	0.010 (0.008)	0.003 (0.052)	0.000 (0.767)	−0.001 (1.303)
Year	3.507 *** (0.370)	0.026 *** (0.003)	0.026 *** (0.002)	0.026 *** (0.002)	0.022 (0.016)	0.020 (0.234)	0.018 (0.397)

*** and * indicate significant at 1% and 10% levels, respectively. Standard errors are in parenthesis.

**Table 6 ijerph-19-10629-t006:** PVAR style GMM results.

LNGDPpc		North			South		
	lngdppc L1.	0.853 ***	0.020	0.000	2.111 ***	0.233	0.000
	lngex L1.	0.038 ***	0.007	0.000	−0.592 ***	0.114	0.000
	lnco_2_ L1.	−0.007	0.020	0.724	−0.533 ***	0.133	0.000
	lnff L1.	0.278 ***	0.041	0.000	−0.058	0.075	0.442
	lnfdi L1.	0.021 ***	0.003	0.000	−0.049 ***	0.009	0.000
lngex							
	lngdppc L1.	−0.978 ***	0.151	0.000	0.285	0.195	0.144
	lngex L1.	1.219 ***	0.055	0.000	0.704	0.128	0.000
	lnco_2_ L1.	0.283	0.166	0.089	−0.198	0.112	0.076
	lnff L1.	1.776 ***	0.405	0.000	0.003	0.095	0.972
	lnfdi L1.	0.065 **	0.020	0.001	0.017	0.009	0.070
lnco_2_							
	lngdppc L1.	−0.122 *	0.060	0.044	4.418	0.947	0.000
	lngex L1.	0.042 *	0.019	0.026	−2.472	0.442	0.000
	lnco_2_ L1.	0.945 ***	0.050	0.000	−1.136	0.528	0.032
	lnff L1.	0.241 *	0.099	0.015	−0.379	0.298	0.204
	lnfdi L1.	0.018 **	0.006	0.001	−0.164	0.042	0.000
lnff							
	lngdppc L1.	0.048	0.040	0.233	0.709	0.168	0.000
	lngex	−0.070 ***	0.009	0.000	−0.365	0.082	0.000
	lnco_2_ L1.	0.101 **	0.035	0.003	−0.317	0.094	0.001
	lnff L1.	0.289 ***	0.067	0.000	0.759	0.062	0.000
	lnfdi L1.	−0.013 ***	0.003	0.000	−0.028	0.009	0.002
lnfdi							
	lngdppc L1.	−2.102 **	0.823	0.011	17.475	3.713	0.000
	lngex L1.	1.870 ***	0.293	0.000	−9.639	1.885	0.000
	Lnco_2_ L1.	−1.681 **	0.765	0.028	−8.022	2.099	0.000
	lnff L1.	15.946 ***	1.676	0.000	−0.884	1.381	0.522
	lnfdi L1.	0.903 ***	0.155	0.000	−0.181	0.193	0.348

***, ** and * indicate significant at 1%, 5% and 10% levels, respectively.

**Table 7 ijerph-19-10629-t007:** Variance decomposition results.

LNGDPpc											
	1	1.000	0.000	0.000	0.000	0.000	1.000	0.000	0.000	0.000	0.000
	2	0.842	0.024	0.007	0.006	0.121	0.723	0.074	0.157	0.000	0.046
	3	0.678	0.055	0.025	0.051	0.192	0.795	0.057	0.112	0.000	0.036
	4	0.539	0.082	0.055	0.114	0.209	0.718	0.064	0.163	0.000	0.055
	5	0.430	0.105	0.097	0.170	0.197	0.751	0.059	0.140	0.001	0.050
	6	0.348	0.123	0.149	0.207	0.173	0.704	0.058	0.176	0.001	0.061
	7	0.285	0.136	0.208	0.223	0.147	0.721	0.058	0.163	0.001	0.058
	8	0.237	0.147	0.270	0.224	0.122	0.687	0.055	0.193	0.001	0.064
	9	0.200	0.154	0.331	0.213	0.101	0.693	0.058	0.185	0.002	0.062
	10	0.170	0.160	0.390	0.197	0.083	0.665	0.054	0.212	0.002	0.067
lngex											
		0.000	0.000	0.000	0.000	0.000	0.013	0.987	0.000	0.000	0.000
	1	0.173	0.827	0.000	0.000	0.000	0.034	0.927	0.028	0.000	0.012
	2	0.190	0.758	0.019	0.015	0.017	0.028	0.902	0.037	0.000	0.033
	3	0.177	0.686	0.059	0.048	0.031	0.048	0.853	0.053	0.001	0.045
	4	0.152	0.621	0.113	0.080	0.035	0.045	0.838	0.056	0.001	0.060
	5	0.126	0.566	0.174	0.102	0.032	0.058	0.811	0.061	0.002	0.068
	6	0.104	0.519	0.239	0.112	0.026	0.057	0.802	0.060	0.003	0.078
	7	0.086	0.479	0.302	0.112	0.021	0.065	0.786	0.061	0.004	0.084
	8	0.072	0.445	0.362	0.106	0.016	0.065	0.780	0.059	0.005	0.091
	9	0.060	0.415	0.416	0.096	0.013	0.071	0.769	0.058	0.006	0.096
	10	0.051	0.389	0.463	0.086	0.010	0.013	0.987	0.000	0.000	0.000
lnco_2_											
		0.000	0.000	0.000	0.000	0.000	0.942	0.000	0.057	0.000	0.000
	1	0.115	0.002	0.884	0.000	0.000	0.763	0.105	0.094	0.001	0.038
	2	0.153	0.002	0.833	0.001	0.012	0.816	0.075	0.081	0.000	0.028
	3	0.171	0.008	0.785	0.007	0.028	0.762	0.099	0.096	0.001	0.043
	4	0.171	0.019	0.749	0.022	0.039	0.790	0.085	0.087	0.000	0.038
	5	0.160	0.031	0.725	0.040	0.044	0.759	0.096	0.099	0.000	0.047
	6	0.145	0.042	0.713	0.056	0.044	0.777	0.088	0.091	0.000	0.044
	7	0.129	0.053	0.712	0.066	0.040	0.755	0.092	0.102	0.000	0.050
	8	0.115	0.063	0.717	0.071	0.035	0.767	0.088	0.097	0.000	0.048
	9	0.102	0.072	0.725	0.071	0.030	0.751	0.089	0.107	0.000	0.053
	10	0.090	0.079	0.735	0.069	0.026	0.942	0.000	0.057	0.000	0.000
lnff											
		0.000	0.000	0.000	0.000	0.000	0.699	0.001	0.069	0.230	0.000
	1	0.111	0.123	0.109	0.656	0.000	0.623	0.074	0.075	0.194	0.034
	2	0.155	0.216	0.117	0.486	0.026	0.664	0.056	0.073	0.181	0.025
	3	0.167	0.309	0.100	0.370	0.054	0.648	0.065	0.081	0.169	0.037
	4	0.154	0.379	0.079	0.322	0.065	0.671	0.061	0.076	0.160	0.032
	5	0.134	0.425	0.072	0.306	0.063	0.663	0.062	0.086	0.152	0.038
	6	0.114	0.453	0.084	0.295	0.054	0.677	0.061	0.081	0.146	0.035
	7	0.097	0.466	0.114	0.277	0.046	0.671	0.060	0.090	0.140	0.038
	8	0.084	0.468	0.156	0.253	0.040	0.680	0.062	0.086	0.136	0.036
	9	0.072	0.463	0.204	0.226	0.035	0.699	0.001	0.069	0.230	0.000
	10	0.063	0.452	0.254	0.200	0.031	0.623	0.074	0.075	0.194	0.034
lnfdi											
	1	0.461	0.019	0.000	0.093	0.427	0.834	0.001	0.006	0.000	0.158
	2	0.399	0.047	0.005	0.095	0.455	0.768	0.070	0.098	0.000	0.064
	3	0.318	0.069	0.024	0.182	0.406	0.796	0.063	0.089	0.000	0.053
	4	0.260	0.083	0.059	0.251	0.347	0.790	0.065	0.098	0.000	0.046
	5	0.223	0.092	0.106	0.279	0.300	0.799	0.064	0.095	0.000	0.041
	6	0.199	0.098	0.159	0.277	0.268	0.796	0.064	0.099	0.000	0.040
	7	0.181	0.100	0.212	0.262	0.245	0.801	0.065	0.097	0.000	0.037
	8	0.167	0.102	0.261	0.243	0.227	0.799	0.064	0.100	0.000	0.036
	9	0.155	0.103	0.304	0.225	0.213	0.802	0.065	0.098	0.001	0.035
	10	0.144	0.104	0.342	0.210	0.200	0.800	0.064	0.101	0.000	0.035

**Table 8 ijerph-19-10629-t008:** Granger causality.

	North				South			
Null Hypothesis	W-Stat.	Zbar-Stat.	Prob.	Direction of Causality	W-Stat.	Zbar-Stat.	Prob.	Direction of Causality
LNGEX ↔ LNGDPpc LNGDPpc ↔ LNGEX	3.499 5.23 *	0.758 2.075	0.448 0.038	Uni-directional	4.766 * 7.789	1.858 4.336	0.063 1.000	Uni-directional
LNCO_2_ ↔ LNGDPpc LNGDPpc ↔ LNCO_2_	9.745 5.869 *	5.498 2.557	4.000 0.010	Uni-directional	8.342 9.375	4.789 5.636	2 × 10^−6^ 2 × 10^−8^	
LNFF ↔ LNGDPpc LNGDPpc ↔ LNFF	9.425 3.802	5.255 0.987	1.000 0.323		5.246 * 4.325	2.251 1.496	0.024 0.134	Uni-directional
LNFDI ↔ LNGDPpc LNGDPpc ↔ LNFDI	4.780 * 2.625	1.730 0.095	0.085 0.924	Uni-directional	8.760 5.050 *	5.131 2.090	3 × 10^−7^ 0.036	Uni-directional
LNCO_2_ ↔ LNGEX LNGEX ↔ LNCO_2_	5.831 * 1.804	2.528 −0.528	0.014 0.597	Uni-directional	5.827 ** 2.567	2.727 0.055	0.006 0.955	Uni-directional
LNFF ↔ LNGEX LNGEX ↔ LNFF	6.162 ** 2.648	2.779 0.109	0.005 0.912	Uni-directional	3.027 2.812	0.432 0.256	0.665 0.797	
LNFDI ↔ LNGEX LNGEX ↔ LNFDI	1.522 * 2.596	−0.741 0.073	0.458 0.943	Uni-directional	3.891 6.655 ***	1.140 3.406	0.254 0.000	Uni-directional
LNFF ↔ LNCO_2_ LNCO_2_ ↔ LNFF	4.939 3.472	1.851 0.737	0.061 0.466		3.047 4.182	0.448 1.378	0.653 0.167	
LNFDI ↔ LNCO_2_ LNCO_2_ ↔ LNFDI	2.535 1.533	0.027 −0.733	0.974 0.463		3.970 3.252	1.205 0.616	0.228 0.537	
LNFDI ↔ LNFF LNFF ↔ LNFDI	12.899 3.2932	7.892 0.602	3.000 0.547		5.246 ** 4.325	2.251 1.496	0.024 0.134	Uni-directional

Note: ***, **, * indicates 1%, 5%, and 10% level of significance, respectively.

## Data Availability

World Bank Group. (2020). World Development Indicators 2020, Washington DC: World Bank. [Dataset for GDP, FDI]. https://datacatalog.worldbank.org/dataset/world-development-indicators (accessed on 23 October 2021).

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
