# Peer review of "Economic Growth and Environmental Quality: Analysis of Government Expenditure and the Causal Effect"

_ijerph, 2022, doi:10.3390/ijerph191710629_

Round 1
Reviewer 1 Report
Dear authors, please see below some comments undertaken with the hope of helping you to improve your interesting paper. The paper has the virtue of being based in an innovative and sound statistical work that, as the authors claim, goes further and improves what has been done in the context of African countries.
But unfortunately, in my opinion, this empirical work is framed wrongly in a failed literature on the complex relationship between EG and EQ, which is the EKC literature. So first please see some comments on why EKC literature I think is not a sound frame to insert your work. Then, some critics on the selection of the dependent variable to refer EQ, and its implications. And finally, some short suggestions on how to proceed to carry your research effort to a good port.
1. On EKC.
EKC literature boomed over the early 90s as an apparent and stylised explanation of the complex relationship between economic growth and environmental quality. Authors looked for a general theory, able to give account of these variables’ relationship everywhere at every time. Thirty years later, thousands of scientific papers have returned this theory to a much lesser relevant position. Even more, I would say it has fallen into irrelevance. Many authors assumed the task of proving that that relationship, if so, was partial and more casual than causal. And, of course, its chief implication, that growth by itself lead to a better environmental quality, is completely indefensible, in light of the present climate, ecological and biodiversity crisis.
Effectively, several assert the importance of the territorial scale and how global economic integration proved EKC for countries/regions specialised in high value and more clean processes of the whole value chain while refusing it for those holding the lesser valued and dirty parts of those processes. Others proved EKB incompatible with fields of the environment exhibiting cumulative impacts so that decreasing environmental damages per economic unit may lead to lesser stocks of environmental quality (Ranjan and Shortle, 2007). In the light of global climatic and biodiversity emergency, could we, in any sense, claim EQ is improving with global economic growth? Furthermore, literature on defensive expenditure proved that in presence of environmental damages and increasing social preferences for EQ, the increase of EX needed to keep social welfare constant makes the presumed positive relationship between EG and EQ a fallacy. It pushed forward since the same 90s a movement to filter growth indicators, i.e. GDP from its spurious content of expenditure that add nothing to social welfare. Also, it is in the ground of the efforts to develop new indicators to assess the societies’ quality of life that have had relevant scholars busy over the last couple of decades, including some Nobel prizes (see, particularly, Mis-measuring our lives by Stiglitz, Sen and Fitoussi.
In the context of the African continent, what could be said about? Guinea Equatorial should be experiencing a phase of increasing EQ due its high level of per capita income in the continent. Yet, in a context of increasing social inequality, the majority of the population, still very poor, should be demanding private and public goods in a way in which the resultant EQ should be decreasing. On the other hand, the current technological revolution of the renewable source energies would be allowing very poor areas of, for example, rural Senegal (and of many other African countries) reaching rural electrification through a clean technology instead of by adopting a dirtier one. The greener technology would be also the cheaper one. Even more, renewable based electrification in rural areas would be cheaper than continue harvesting mangrove or any other wild forest wood, if putting some basic price to the increasing time communities devote to gather wood as local energy vector. In many aspects, EKC should be tunnelled, rather than run
To some extent, EKC incongruency rise from the very roots of the economic theory lying under it. Natural and physical capital are seen as substitutive types of capital with an increasing marginal rate of substitution. However, many authors have proved complementarity as ruling the relationship between both forms of capital. In many places of Africa where a fatal combination of a global human impact (climate change with its consequences of increasing droughts and floods) and poorly planed human interventions (i.e. dams damaging the ecological flow of rivers), additional physical capital only would lead to increase food production if, at the same time, the degraded capital natural were substantially improved. Complementary doses of natural, man-made and human capital would be required to advance in economic growth (and welfare) even from very early stages of EG.
The above referred ones are single examples of why generalizations may give place to conclusions that would support policies that mismatch the common sense inspired on the empirical observation. In the context of Africa, EKC may be signalled as wrong, irrelevant, and misguided. It does not accomplish most of the time; many results are at the best irrelevant; and policies supported on it would become misguided.
2. The variable to refer the construct of EQ.
Using CO2 emissions as a proxy of EQ does not make sense, in my opinion. Precisely CO2 emissions, a global pollutant related to greenhouse effect and hence global warming, cannot be expressive of the EQ of any particular country. The particular relationship between this environmental externality and the EQ goes through the aggregate effect of GHG emissions on biodiversity, droughts, forest fires, etc. So it evolves with regardless on the particular country’s/region’s CO2 emissions.
The paper just evidence whether or not there is a process of relative decoupling between GDP per capita and C02 emissions, that’s to say, it is an indicator of the relative decarbonisation of the NASA countries at the two considered sub-regions. But still I think that it would not very relevant in the general context of the set of environmental variables that could be expressive of the real environmental decadency that many African countries are experiencing. What is going on with the CO2 sinks in many African countries would be another aspect to take into account, together with emissions. The combination of droughts and wild forest fires would be releasing C02 in net terms, affecting significantly to the carbon balance of the referred countries and regions.
3. What, in my opinion, could be done?
To save the imponent statistical work and give some meaning to the obtained results, the simplest solution would be rewriting the theoretical framework, leaving aside EKC theory. The relationship between EG and EQ mediated by GEX, extraordinary relevant, would be rescued for a newly defined framework, probably with different policy implications.
Please, consider the following papers related to EKC criticism:
Kaika, D., & Zervas, E. (2013). The environmental Kuznets curve (EKC) theory. Part B: Critical issues. Energy Policy, 62, 1403-1411.
Ranjan R., Shortle J. The environmental Kuznets curve when the environment exhibits hysteresis. Ecological Economics, 64 (1) (2007), pp. 204-215.
Caviglia-Harris, J. L., Chambers, D., & Kahn, J. R. (2009). Taking the “U” out of Kuznets: A comprehensive analysis of the EKC and environmental degradation. Ecological Economics, 68(4), 1149-1159.
Dogan, E., & Inglesi-Lotz, R. (2020). The impact of economic structure to the environmental Kuznets curve (EKC) hypothesis: evidence from European countries. Environmental science and pollution research, 27(11), 12717-12724. LA CONFIRMA PARA EUROPA.
Gill, A. R., Viswanathan, K. K., & Hassan, S. (2018). The Environmental Kuznets Curve (EKC) and the environmental problem of the day. Renewable and sustainable energy reviews, 81, 1636-1642.
Karsch, N. M. (2019). Examining the validity of the environmental Kuznets curve. Consilience, (21), 32-50. “…there is no guarantee that economic growth will lead to an improved environment. More specifically, this essay proves that the notion that income growth by itself will be beneficial for the environment is fictitious, as a causal relationship between income and environmental quality cannot be consistently demonstrated”.
Perman, R., & Stern, D. I. (2003). Evidence from panel unit root and cointegration tests that the environmental Kuznets curve does not exist. Australian Journal of Agricultural and Resource Economics, 47(3), 325-347.
***Stern, D. I. (2003). International society for ecological economics internet encyclopaedia of ecological economics the environmental Kuznets curve. Department of Economics, Rensselaer Polytechnic Institute.
A number of critical surveys of the EKC literature have been published (e.g. Ansuategi et al., 1998; Arrow et al., 1995; Ekins, 1997; Pearson, 1994; Stern et al., 1996; Stern, 1998.
Stern et al. (1996) raised the issue of heteroskedasticity that may be important in the context of cross-sectional regressions of grouped data (see Maddala, 1977). Schmalensee et al. (1998)
The EKC is an essentially empirical phenomenon, but most of the EKC literature is econometrically weak. It is very easy to do bad econometrics and the history of the EKC exemplifies what can go wrong. The EKC idea rose to prominence because few paid sufficient attention to econometric diagnostic statistics. Little or no attention has been paid to the statistical properties of the data used such as serial dependence or stochastic trends in time series and few tests of model adequacy have been carried out or presented. However, one of the main purposes of doing econometrics is to test which apparent relationships, or "stylized facts", are valid and which are spurious correlations.
…the course of economic development. If there were no change in the structure or technology of the economy, pure growth in the scale of the economy would result in a proportional growth in pollution and other environmental impacts. This is called the scale effect. The traditional view that economic development and environmental quality are conflicting goals reflects the scale effect alone. Proponents of the EKC hypothesis argue that “at higher levels of development, structural change towards information-intensive industries and services, coupled with increased environmental awareness, enforcement of environmental regulations, better technology and higher environmental expenditures, result in leveling off and gradual decline of environmental degradation.” (Panayotou, 1993, p 1).
Specific final comments
Kapital intensity and environmental damage. Pg 2, lines 61 et seq. Wood-based domestic energy (labour intensive) vs rural electrification with distributed photovoltaic (capital intensive).
The lack of urgency makes ignores environmental in governmental expenditure (GEX). P2, lines 68 et seq. This perspective is changing quickly due to the environmental crisis in sub-Saharan Africa.
P2, lines 73 et seq. Basic and luxury public goods. They are complementary!!! EQ critically influence water availability, health conditions, food security…
As GEX increases, EQ is given more consideration.
Economic growth and social progress in developing countries.
P2, lines 92 et seq. Pollutant as CO2 are produced during the industrial production and are one of the most significant causes of environmental damage.
P3, lines 10-4 et seq. The mediation of GEX between EG and EQ is not well described. Scale versus innovation effect.
P3, lines 11-6 et seq. FDI meaning.
Best wishes.
Author Response
All contributions and suggestion raised were considered in order to improve on the quality of the study

Reviewer 2 Report
Dear Authors:
I am glad to review your article and provide my comments as following:
In abstract, please use quartiles instated quantiles since quartiles are most common in statistical world. In addition, abbreviations in the line 21 should be addressed. For instance, LNGEX, LNGDPppc, etc.
Subscript should be properly used. Such as CO2 instead of CO2 in line 18. Some of abbreviation should be introduced, such as EKC in line 35, UNFCC in line 73, NASA in line 101, FDI in line 287.
In line 130, you started to introduce the objectives of this study. But the related literature is not enough. You may need to combine the literature review sections into the introduction. Then, you may need to have a short, comprehensive and clear objectives of your study at end of the introduction part.
Table 1. Data set for what? There are factors, including EG, GFE, EQ and FDI with indications in Table 1. Why do you include the FDI?
In Table 2, descriptive statistics for what? The abbreviations at first row of Table 2 should be introduced as note. Where are the data sets to calculate mean, median, max. and minimum?
Table 3, there is p-value. How do you use this p-value from the correlation test? Please state that statistical significance or p-value is 5% (or other values) for your study. Please read and cite these articles:
Qian, X., Chen, G., Kattel, B., Lee, S., & Yang, Y. (2018). Factorial analysis of vertical ground reaction force and required coefficient of friction for safety of stair ascent and descent. Inter. J. Ind. Oper. Res, 1(002).
Rice, W. R. (1990). A consensus combined P-value test and the family-wide significance of component tests. Biometrics, 303-308.
The fonts and formats of the equations should be changed to the same style.
In Figures 1 and 2, what is the x-axis? how did you get these results? simulation?
In line 453, Fig. 2 or Fig. 3?
In Table 5, Quantile regression? what's the meaning of these data? you need to add some more description on the results. What is the number in the bracket?
Poly implication in the conclusion section should be added as discussion part.
You did a lots of analysis on the north and south. Based on your analysis, your major results or conclusion are not very clear in either conclusion or abstract.
In conclusion, you should avoid to use references (in line 594). You need to make unique conclusion based on your analysis and results.
I am looking forward to see the revised version.
Best regards,
Reviewer
Author Response
All suggestion and contributions raised were considers by authors and have made the necessary change

Reviewer 3 Report
Firstly, I would like to congratulate the authors for the article “Economic Growth and Environmental Quality: Analysis of Government Expenditure and the Causal Effect”.
I believe there is an error in the acronym used in the abstract panel quantile regression (QPR), after which the authors use PRQ. Still about the acronyms, the authors do not define GMM, WWII, etc. The authors present several acronyms, and there is some confusion so that the reader can understand all of them. I strongly recommend more care in the presentation of acronyms, perhaps rethinking the relationship between terms and acronyms.
The relationships presented in the abstract must also be revised. I believe that more important than relationships is what they mean and what their impacts are. Authors should think more about the contributions that the research findings have made.
In the introduction, the problem of acronyms remains, for example, the authors should present the meaning of EKC.
In general the text needs some corrections, eg page 3, line 121: Baz et al, Gani and Hadj [37–39]. The correct one would be Baz et al. [37], and Gani and Hadj [39]. Several other minor errors in the text are evident.
Line 124: studies by [40–43]
Lines 129: whiles [17, 45] consider MENA and NA, respectively....
Line 232: Naz et al, Shoaib et al, Sunkanmi et al [42, 53, 54]
Line 241: Isik et al [9, 57]
Again about acronyms, perhaps a list of acronyms at the beginning of the article would help the reader understand the text better. This is clear if it is in accordance with the journal's writing and formatting rules.
This statement is a bit bold for this study: “This research is quite unique in various 96 ways as:”. In addition, it is poorly positioned in terms of text writing, as it is not a topic in the sequence.
Regarding the objective of the article, I strongly recommend the authors to rethink their presentation. There is one objective statement in the abstract that is not very clear, and another is the introduction. Authors should think about clarity and conciseness to present the general objective of the article. The objective should be the same in both sections, without variations.
Abstract: “this review looked at the impact of economic growth (EG) on EQ, as well as the role of government finance expenditure (GEX) in Northern and Southern Africa (NASA) republics from 2000 to 2016.”
Introduction: “… study looks at the role of GEX in EG-EQ in NASA economies.”
Another point that is not clear and that ends up presenting the text with some ambiguity is the research method. At various times the authors give the impression that they have carried out a review. Other parts of the text demonstrate an empirical research intent. It is confusing for the reader.
I recommend rethinking titles that use synopsis: Synopsis of EQ-GEX.... Here the authors present the theoretical assumptions that underlie the constructs of the article. Perhaps a title more consistent with this type of content.
Some paragraphs are a bit long and poorly referenced. Some parts of the text need to be corrected and articulated better.
In section 3, the authors present results, but they should only present the materials and methods. This section should focus on the methodological procedures adopted and applied. Justify the methodological choices and leave all the results for the next section: presentation of the results. In my opinion the confusion starts at line 309.
Explain why all variables in table 2 start with LN.
Authors should present the analysis parameters in tables 2, 3 and 4, so that the reader will better understand the results presented. Some comment on the results is also required for clarification.
It is necessary to explain the information in Figure 4, as in the others. Some comment that allows understanding the relationships between the results. In addition, it needs to improve the readability of the figure. Some parts of the text are difficult to understand.
I missed an analysis to test the behavior of the data in the two groups, so test if they in fact do not have commonality.
I believe that a suggestion that could be accepted is to present a model to be tested at the end of the theoretical reference section. Raise some hypotheses and present a discussion after the results. Here we have some information that can help to understand the phenomenon studied. The authors demonstrate a negative relationship between CO2 generation and government investments, but some intervening variables that should be pointed out are evident. Even if the authors do not explore these variables, it would be only fair for this discussion to present this general picture to the reader.
The authors present some graphic elements without due care. Figure 7 needs to be explained and explored. The same goes for others.
In the introduction, the authors talk about the great differentials of this article, but in the end the deliverables are not evident. It is necessary to establish a discussion with the theoretical framework presented. In the conclusion, a discussion on the advances is expected. Not just exploring what has already been found in the analyses. Authors must clearly present limitations and proposals for future studies. We can find several limitations, including the reductionist perspective of the relationship between the variables. It's not a problem, but it should be pointed out.
Author Response
Thank you so much for you input, suggestion and contribution and all were considered

Round 2
Reviewer 1 Report
1. You cannot say this: “In the developmental state of every economy, however, after reaching a particular economic level, environmental pollution decreases due to the deployment of environmentally friendly technology, the expansion of the information and service industries, and high demand in environmental protection, which is induced through environmental expenditure (EX)”, without making a big mistake, that should not be published nowadays in a scientific journal.
2. In a globalised economy, you should not split a production process by country, but take into account the emissions associated to the whole production process (life cycle approach). The ability of developed economies to shift dirty process of production to developing countries do not make them cleaner, indeed.
3. CO2 emissions is not a proper indicator for EQ. The impact of emissions on EQ is better approached by CO2 concentration indicator, as it gives better account for global warming and the extreme climate phenomena we all know as climate change. Climate change is a cumulative process directly depending from concentration of CO2 eq., not from emissions. Please, see the following data:
|
|
CO2 emissions t |
CO2 concentration ppm |
GDP World trillion $ contant 2017 |
||
|
Year 1990 |
20.625.273 |
350 |
51,4 |
|
|
|
Year 2019 |
34.344.006 |
460 |
130,42 |
|
|
|
2019/1990 |
1,665 |
1,314 |
2,537 |
|
|
Globally, economy increases faster than emissions, but concentration still rise, so all countries are now worse off.
4. Using CO2 concentration indicator, there are not countries experiencing improvements with economic growth. In all countries, although sound metrics are still to be developed, fragmented information points to climate change related costs are higher than benefits. Sorry, but EKC doesn’t work in this case (and many others). So, please, you cannot make a general statement as you did in the above paragraph.
5. Regarding African countries, I think that there is no a single one, neither at the north nor at sub-Saharan region, experiencing a positive net effect on EQ (which is different to experience lesser emissions), derived from climate change.
6. Still, I think your work is very interesting, but it would gain credibility and utility, de coupling it from EKC. Finally, if you are convinced EKC because you love it, I still will be favourable your paper to be published.
Have a nice time.
Author Response
we are grateful for your suggestions and contribution which have help us improve on our paper. Authors see most of your contribution well deep and will help us a lot in our next research work

Reviewer 2 Report
Dear Authors:
I found that you made changes and improvements based on the previous comments. I have minor comments to be addressed:
End of the literature review at page 6, you need to summarize and emphasize the research gaps and objective of your study in 1-2 sentences.
Format and fonts of the equations should be addressed.
Figure 5 and Figure 6 should be improved with clear subtitles on the each small sub-diagrams.
Description of Figure 7 is missing. Please add it.
For the P-value and discussion of the significance on the factors and response on Table 3, please cite the following suggested articles:
Factorial analysis of vertical ground reaction force and required coefficient of friction for safety of stair ascent and descent. Inter. J. Ind. Oper. Res, 1(002).
A consensus combined P-value test and the family-wide significance of component tests. Biometrics, 303-308.
In conclusion, authors only summarized major results. What's your conclusion from these results? What's the novelty of your study as well as future study?
Policy implication and limitation of study should be added into the discussion part.
Please check the writing styles, spelling and grammar with professorial writer.
I hope you will address these points to further improve the quality of your manuscript.
Best regards,
Reviewer
Author Response
your suggestions raised have come a long way in improving you research work

Reviewer 3 Report
The authors made some good revisions to the manuscript. However, it still needs at least a few points of improvement.
The methods section should be made clearer, including I strongly recommend authors to separate what is results from methodological procedures.
Authors must review the manuscript well, eliminating citation and writing errors. It is necessary to establish some discussion in the article after the presentation of the results.
Graphic elements (tables and figures) must ensure legibility and understanding. Authors should provide some explanation or discussion after the graphics.
I still believe that the authors can bring in the abstract and in the conclusion the most important elements related to the research findings. It is necessary to demonstrate where the article advances and/or points out ways.
Author Response
Your contribution and suggestions have helped us to improve on paper and we are grateful
